# Movement Quality Parameters during Gait Assessed by a Single Accelerometer in Subjects with Osteoarthritis and Following Total Joint Arthroplasty

**DOI:** 10.3390/s22082955

**Published:** 2022-04-12

**Authors:** Jill Emmerzaal, Kristoff Corten, Rob van der Straaten, Liesbet De Baets, Sam Van Rossom, Annick Timmermans, Ilse Jonkers, Benedicte Vanwanseele

**Affiliations:** 1Human Movement Biomechanics Research Group, Department of Movement Sciences, KU Leuven, 3001 Leuven, Belgium; sam.vanrossom@kuleuven.be (S.V.R.); ilse.jonkers@kuleuven.be (I.J.); 2REVAL Rehabilitation Research Centre, Faculty of Rehabilitation Sciences, Hasselt University, 3590 Diepenbeek, Belgium; rob.vanderstraaten@uhasselt.be (R.v.d.S.); annick.timmermans@uhasselt.be (A.T.); 3Department of Orthopaedics, Ziekenhuis Oost Limburg, 3600 Genk, Belgium; kristoff.corten@zol.be; 4Pain in Motion Research Group (PAIN), Department of Physiotherapy, Human Physiology and Anatomy, Vrije Universiteit Brussel, 1050 Brussel, Belgium; liesbet.de.baets@vub.be

**Keywords:** biomechanics, trunk-worn accelerometer, movement quality, hip osteoarthritis, knee osteoarthritis, total knee arthroplasty, level walking

## Abstract

This study’s aim is threefold: (I) Evaluate movement quality parameters of gait in people with hip or knee osteoarthritis (OA) compared to asymptomatic controls from a single trunk-worn 3D accelerometer. (II) Evaluate the sensitivity of these parameters to capture changes at 6-weeks, 3-, 6-, and 12-months following total knee arthroplasty (TKA). (III) Investigate whether observed changes in movement quality from 6-weeks and 12-months post-TKA relates to changes in patient-reported outcome measures (PROMs). We invited 20 asymptomatic controls, 20 people with hip OA, 18 people pre- and post-TKA to our movement lap. They wore a single trunk-worn accelerometer and walked at a self-selected speed. Movement quality parameters (symmetry, complexity, smoothness, and dynamic stability) were calculated from the 3D acceleration signal. Between groups and between timepoints comparisons were made, and changes in movement quality were correlated with PROMs. We found significant differences in symmetry and stability in both OA groups. Post-TKA, most parameters reflected an initial decrease in movement quality at 6-weeks post-TKA, which mostly normalised 6-months post-TKA. Finally, improved movement quality relates to improvements in PROMs. Thus, a single accelerometer can characterise movement quality in both OA groups and post-TKA. The correlation shows the potential to monitor movement quality in a clinical setting to inform objective, data-driven personalised rehabilitation.

## 1. Introduction

Hip and knee osteoarthritis (OA) are frequent, disabling musculoskeletal disorders and are a leading cause of lower extremity disabilities [1]. Symptomatic OA is characterised by pain, stiffness, physical disabilities, and difficulties in performing daily life activities [1]. Patient-reported outcome measures (PROM) are most frequently used to measure the impact on everyday functioning but lack biomechanical insights [2]. However, PROMS are prone to subjective factor, whereas we need objective measures to inform rehabilitation that ideally relate to biomechanical function, as this is what we aim to remediate. Thus, knowing about the underlying mechanism of a poorer outcome can help identify more personalised treatment targets.

Laboratory studies have identified biomechanical changes in the gait pattern of patients with hip or knee OA [3,4,5,6]. The normalisation of lower limb biomechanics is often strived for, particularly in the case of joint replacement surgery. This evaluation can be done with either highly simple (step length) or highly complex (up to the level of loading) parameters. Research has shown that an altered gait pattern can still be present up to one year following total knee arthroplasty (TKA) or total hip arthroplasty (THA) [7,8,9,10,11]. Therefore, measuring biomechanical alterations in a clinical setting would be beneficial to monitor the gait pattern. However, to date, conventional methods to monitor and evaluate the gait pattern relies on extensive lab-based motion capture systems that do not necessarily reflect movement patterns outside of a lab.

Alternatively, accelerometers bear the advantage of easy mounting to the patients without the restriction or complexity of a gait lab [12]. A scoping review by Kobsar et al. (2020) demonstrated the increased popularity of using wearable technology for (mostly spatio-temporal) gait analysis in people with OA [13]. More precise, a single accelerometer at the lower trunk level was used as an easy and unobtrusive method to assess gait- and running biomechanics, both in healthy adults and patients with neuromuscular disorders [12]. The use of only one sensor allows (as a proxy to) objectively detecting changes in the three-dimensional movement characteristics of the centre of mass (CoM) [14,15], providing global kinematical information during walking. As such, three different components of gait can be assessed, i.e., vertical (VT) acceleration reflecting up and down movement, mediolateral (ML) acceleration reflecting left-right sway, and thus stability and anterior–posterior (AP) acceleration reflecting propulsion and braking.

Dedicated accelerometer-derived measures, both in the time or frequency domain, have been used to assess movement quality in terms of symmetry, complexity, smoothness, and stability. Previously, researchers used these parameters to determine the variability and knee function in anterior cruciate ligament deficient knees [16,17,18], gait smoothness in older adults with central dysfunction [19], fall risk in older adults [20,21], and motor recovery after stroke [22], and Parkinson’s disease [23]. Furthermore, these parameters were sensitive enough to discriminate between the gait pattern of healthy controls and pre-manifested individuals with Huntington’s (i.e., gene carriers, but not displaying any functional decline) [24] and early gait alteration in patients with multiple-sclerosis [25]. A healthy physiological and biomechanical function is characterised by high complexity that enables adaptability to unpredicted events in daily life [26], an optimal level of stability that attenuates small perturbations and transitions to different motor patterns, and a smooth motor pattern that indicates well-trained motor behaviour [22]. Therefore, these parameters could be instrumental in assessing the functionality of a patient in a clinical setting and ideally could objectively inform targeted rehabilitation strategies. However, the discriminant ability of the parameters in people with OA from asymptomatic controls is still mostly unexplored, thereby exploring if measured differences represent actual differences that surpass measurement error and, thus, related to the OA status. Furthermore, it would be interesting to assess whether these parameters can monitor the impact of a total joint arthroplasty on gait characteristics and the follow-up of the recovery process. The latter is needed to obtain better insights into the typical evolution of the movement quality post-TKA. Furthermore, the sensitivity of a parameter to change is essential for monitoring and diagnostic work. Thus, providing a step towards data-informed personalised rehabilitation by showing potential rehabilitation targets that are easy to monitor in a clinical setting.

Therefore, we have a threefold aim: (1) to evaluate which parameters derived from a single trunk worn accelerometer can distinguish the gait pattern of people suffering from symptomatic hip or knee OA compared to asymptomatic controls; (2) to evaluate the sensitivity of these parameters to capture changes in gait properties at six weeks, three, six, and 12 months following total knee arthroplasty (TKA); and (3) to investigate whether changes in movement quality parameters between 6 weeks and 12 months following TKA are related to patient-reported functioning, pain, symptoms, sports/recreation, and quality of life.

## 2. Materials and Methods

### 2.1. Study Design and Study Sample

The local ethics committees of the University Hospital Leuven, in collaboration with Ziekenhuis Oost-Limburg (Genk, Belgium) and Jessa Hospital (Hasselt, Belgium), approved this prospective cohort study (S59857). The cross-sectional analysis consisted of 20 people with unilateral end-stage hip OA, 18 people with unilateral end-stage knee OA and 20 asymptomatic controls. The inclusion and exclusion criteria are described in Table 1. The eighteen people with knee OA were treated with a TKA, and 17 were re-evaluated at six weeks, three months, six, and 12 months postoperatively. One participant dropped out due to a herniated disc with functional impairment. This study is a secondary analysis of a larger project (S59857) that evaluated the hip and knee joint contact forces in people with degenerative joint disorders and following a total joint arthroplasty. The sample size was based on the compartmental forces measured in subjects with an instrumental knee prosthesis (1.61 ± 0.305 body weight during gait [27]). Assuming that a change in contact forces of one standard deviation is significant and to achieve a power of 0.8, a sample of 14 subjects was required. Taking a possible 15–20% loss of participants into account during the follow-up, we recruited 18 to 20 participants per cohort.

### 2.2. Data Acquisition

We placed a single tri-axial Inertial Measurement Unit (IMU) (MVN BIOMECH Awinda, Xsens Technologies, sampling at 60 Hz [28]) with a 3D accelerometer at the level of L5/S1 using double-sided tape. We used an additional Velcro strap around the participant’s waist to further secure the IMU and minimise excessive movement. Subjects were instructed to walk at self-selected speed in a straight line of 10 m across our movement laboratory (MALL, KU Leuven, Belgium) at different evaluation points. People with hip OA were only measured once (pre-THA), and the people with knee OA were evaluated five times (pre-TKA, six weeks, three months, six, and 12 months post-TKA) (see Figure 1). The asymptomatic controls returned to the MALL for a re-evaluation. We used the data from asymptomatic controls to calculate the minimal detectable change of the movement quality parameters by calculating the interclass correlation coefficient (ICC). The Hip disability (Hip OA subjects) and Knee injury Osteoarthritis Outcome Score (Knee OA and Asymptomatic subjects) (HOOS and KOOS) were completed to evaluate patient-reported outcome measures. Figure 1 gives the sensor setup and flow of the data collection including the number of participants measured at each time instance.

### 2.3. Data Processing

For each walking trail, to account for gravity and improper alignment, the sensor tilt was corrected to convert the accelerations from the local sensor XYZ-coordinate system to the global anterior–posterior (AP), vertical (VT), and mediolateral (ML) coordinate system using established methods from Moe-Nilssen et al. 1998 [29]. In short, we used the accelerometer’s capacity as an inclinometer to construct a horizontal and vertical coordinate system [29] by extracting the gravitational components from the signal of each axis, calculating the tilt angles for each axis using trigonometry and finally subtracting the static components from each axis.

After that, the AP and ML accelerations were used to identify the individual left and right steps using methodology adapted from Zijlstra and Hoff (2003) [14]. In short, steps detection was done using the peak of the AP acceleration signal, and the left/right steps were identified with the medio-lateral acceleration signal. Zijlstra and Hoff used three methods for step detection: (1) force plate data, (2) peak AP acceleration, and (3) zero crossing of the AP acceleration signal with a negative slope. They found that the peak AP acceleration closely corresponds to the force plate data. However, finding the appropriate peaks of the AP signal was not easy in some individuals, especially in those with high levels of asymmetry. Therefore, we used the following workflow to detect the target peaks:We determined the average step and stride times by computing the autocorrelation signal from the VT acceleration signal, with the average step and stride time being the first and second dominant peak after zero-phase, respectively;We filtered the AP acceleration signal and detected the maximum peaks. Filter properties: 2nd order Butterworth filter, Cut-off frequency 3 Hz, filtered in both directions;We retained the highest peaks, at least the average step time ±10 samples separated from each other;We imposed that the target peak in the raw AP acceleration signal should fall after the peak from the filtered signal. Therefore, we designed a “relevance window” with the lower bound being the time instance of the peak found in the filtered signal and the upper bound being the time instance of the peak plus 15 samples. Within that relevance window, the correct signal maximum of the AP acceleration was found;All trials were manually checked. The peak finder threshold was manually lowered or increased when errors were spotted (e.g., steps not detected).

To detect left and right steps, we used the ML acceleration signal. Since an accelerometer on the lower trunk can be used as an approximation to the movement of the CoM, inverted pendulum models and experimental data showed that during a stance phase of the left leg, the CoM accelerates to the right (positive ML acceleration). During a stance phase on the right leg, the CoM accelerates to the left (negative ML acceleration). Zijlstra and Hoff showed, again by comparing it to force plate data, that this indeed can be captured by a single accelerometer on the lower trunk with the ML acceleration signal. The steady-state steps [30] were extracted and concatenated to create one long, continuous time series [31]. Considering that the stability and complexity measures are sensitive to the time series length used as input, we used a fixed-step approach to establish the length of the time series [32]. We determined the least number of steps taken by the participants and truncated the signal length of all other participants to that number of steps (n=47, which corresponds to ±1500 samples).

### 2.4. Movement Quality Parameters

Movement quality was then evaluated in terms of (1) movement symmetry, (2) local dynamic stability, (3) movement complexity, and (4) movement smoothness.

First, movement symmetry was quantified as step and stride regularity. These were calculated using the first two dominant peaks after the zero phase of the unbiased autocorrelation with perfect symmetry equal to one [15]. Since a cyclic signal will produce an autocorrelation with peak values with a time lag equivalent to the period of the signal, the first and second dominant peak represents phase shifts equal to one step and one stride, respectively [15]. The unbiased autocorrelation signal was normalised to equal one at zero phase shift. Therefore, the height of the first dominant peak shows the autocorrelation coefficient between consecutive steps, and the height of the second dominant peak shows the autocorrelation coefficient between consecutive strides and is therefore considered a symmetry index. Since ML trunk accelerations produce positive and negative values representing left-right trunk sway, step regularity in the ML direction is always negative. Therefore, the absolute values are used for analysis. In both cases, lower values of the autocorrelation coefficient indicate more asymmetry.

In the second category, local dynamic stability quantified by the maximum Lyapunov Exponent was calculated by estimating the short-term and long-term divergence exponent (LyE $\lambda_{S}$ and LyE $\lambda_{L}$, respectively). The LyE is quantified by calculating the divergence of nearest neighbours in state spaces using Rosenstein’s method [33] and as proposed by Bruijn et al. (2010) [34]. For the calculation of the LyE, we applied time-normalisation, so each stride was 100 samples and set the embedding dimension to 5 [34]. We calculated the time delay as the decrease in the autocorrelation curve of 1−1/e, proposed by Rosenstein and colleagues [33]. LyE is calculated over two time increments: LyE $\lambda_{S}$ over 0–0.5 strides and LyE $\lambda_{L}$ over 4–10 strides. The $\lambda_{S}$ indicates how well the systems deal with perturbations at the step or stride level, indicating gait stability; whereas $\lambda_{L}$ relates the long-range correlations in the gait pattern, thus associated with gait fluctuations [34,35]. Higher values indicate lower dynamic stability [34,36], indicating an unstable gait pattern with a higher risk of falling [36] or more fluctuations in the gait pattern [35], respectively.

The third and fourth categories are the movement complexity and smoothness measure, quantified as sample entropy and log dimensionless jerk (LDLJ-A), respectively. Sample entropy captures waveform predictability with higher values indicating less periodicity, thus, more unpredictability [37]. We used nonlinear mathematical algorithms previously described by Richman and Moorman (2000) [38]. As input for the calculation of the sample entropy, we used the time series sample length (*N*) corresponding to the least number of steps taken as described previously, the series length (m) of 2 data points, and a tolerance window (r) normalised to 0.2 times the standard deviation of the time-series [37].

LDLJ-A assesses movement smoothness by quantifying the changes in the acceleration signal (jerk—a derivative of the acceleration signal) as proposed by Melendez-Calderon et al. (2021) [39]. The Euclidean norm (2-norm) of the acceleration signals (i.e., Pythagorean Theorem over acceleration in VT, ML, and AP direction) was used to calculate the LDLJ-A over each step; thereafter, the average was calculated to obtain a single smoothness measure per subject. A signal that shows minimal changes in the acceleration and deceleration pattern is considered smoother. Lower values indicate a smoother movement pattern [22].

All data were processed and analysed using customised MATLAB scripts (MATLAB 2018b, The Math Works, Inc. Natick, MA, USA).

### 2.5. Statistical Analysis

Data were not normally distributed as assessed by visual inspection of the histogram, Q-Q plot, and Shapiro–Wilk test. Therefore, non-parametric statistics were used. We used the Mann–Whitney U test for group differences between asymptomatic controls and people with hip OA and between asymptomatic controls and people with knee OA. The Minimal Detectable Change (MDC) per dependent variable was calculated using data from the test–retest of all asymptomatic controls using the interclass correlation coefficient (ICC(3,k)) [40]. The MDC was calculated to check whether a difference between the two cohorts is a fundamental difference that surpasses the system’s measurement errors.

Friedman’s chi-square ANOVA was conducted to assess how the parameters evolve after a TKA. When a significant main effect was found (α<0.05), a Wilcoxon signed-rank test with a Bonferroni correction (α<0.005) was calculated to test for differences between timepoints. A Spearman’s ρ correlation coefficient was calculated on the change scores between 6 weeks and 12 months post-TKA to relate changes in movement quality to patient-reported functioning, symptoms, and quality of life. A Spearman’s ρ correlation coefficient between 0–0.25 was considered low, from 0.25–0.5 fair, 0.5–0.75 moderate, and 0.75–1.0 high. Statistical analysis was performed using Python SciPy statistics package (v1.4.1), and missing data were omitted [41].

Results in preoperative cohorts of 20 asymptomatic controls, 18 people with knee OA and 20 people with hip OA. Seventeen people post-TKA were included in the follow-up analysis (one drop-out at 12 months) and in the correlation analysis between movement quality and patient-reported pain, symptoms, ADL, and QOL. The correlation analyses of patient-reported sports/recreation were on 15 people (drop-out of 3 due to an inability to answer the questionnaire).

## 3. Results

### 3.1. Preoperative Cohort Comparison

Subject characteristics are shown in Table 2. Patient-reported outcomes (PROMS) were significantly worse in the OA cohorts than in the healthy controls (Table 2). The PROMS following TKA improved over time, nevertheless they remained significantly lower than the asymptomatic controls (Table 2).

Movement symmetry measured by the step regularity was different between cohorts. People with hip OA had a more asymmetrical gait pattern than asymptomatic controls in all three directions (Figure 2 and Table 3). All these differences exceeded the minimal detectable change threshold (Figure 2). However, people with knee OA also displayed an asymmetrical gait pattern in the VT and ML direction. These differences in people with knee OA exceeded the minimal detectable change (Figure 2).

The long-term divergence exponent (LyE $\lambda_{L}$) was not different in people with hip or knee OA compared to controls (Figure 2 and Table 3). The short-term divergence exponent (LyE $\lambda_{S}$), on the other hand, was significantly lower in people with knee OA in all three directions; and surpassed the minimal detectable change in the AP direction. Additionally, a significantly lower LyE $\lambda_{S}$ in ML direction and bordering on significance level in AP direction was found between people with hip OA and asymptomatic controls, but the difference was too small to surpass the minimal detectable change. No group differences were found for either Sample Entropyor LDLJ-A (Figure 2).

### 3.2. Longitudinal Follow-Up Following TKA

Some changes in asymmetry could be observed during the re-evaluation moments (Figure 3). A trend towards increased step asymmetry was observed in VT and ML direction from pre-TKA to six weeks post-TKA, followed by a significant improvement in step regularity in VT direction at six- and 12-months post-TKA compared to six weeks post-TKA. Step regularity in the ML direction post-TKA remained more asymmetrical than the asymptomatic controls one-year post-TKA.

Short- and long-term divergence exponent did not significantly change over time following TKA. However, we found significantly higher LyE $\Lambda_{L}$ values than the asymptomatic controls at 12-month post-TKA (Figure 3), indicating higher gait fluctuations at that timepoint. Furthermore, the LyE $\Lambda_{S}$ in ML and AP direction was significantly lower than in asymptomatic controls, reflecting a more rigid movement pattern. The LyE $\lambda_{S}$ in the VT direction was no longer significantly different from asymptomatic controls 12 months post-TKA.

There was a reduction in movement complexity (sample entropy) in the VT and ML direction at 6-weeks post-TKA compared to pre-TKA complexity values (Figure 3). Sample entropy in the ML direction was significantly improved at six months post-TKA; however, this improvement was not observed in the VT direction. Movement smoothness quantified as LDLJ-A was not affected by the TKA procedure (Figure 3).

### 3.3. Correlations

Step regularity in the AP direction was faily correlated (ρ=0.48) with quality of life, showing that an increase in step symmetry was correlated with increased patient-reported quality of life (Table 4). The quality-of-life section of the KOOS mainly reflects how well people can “trust” their affected knee (i.e., giving away)—indicating that an increase in step symmetry correlates to the feeling of being able to trust their knee.

Improved stability, both LyE $\lambda_{S}$ and $\lambda_{L}$, were fair to moderately correlated with improved patient-reported pain, quality of life, ADL, and sports and recreation. An increase in LyE $\lambda_{S}$ in ML and AP direction (i.e., a trend towards values found in healthy controls) reflected decreases in pain (ρ=0.51 and ρ=0.54), an increase in the ability to perform daily life activities (ρ=0.43 and ρ=0.55) and quality of life (ρ=0.43 and ρ=0.48). Indicating that a positive change in short term stability (more resembling asymptomatic controls) related to better patient-reported outcome scores. Changes in LyE $\lambda_{L}$ in AP and ML direction were negatively correlated with changes in sports/recreation (ρ=−0.56 and ρ=−0.4), reflecting a decrease in LyE $\lambda_{L}$ towards values measured in healthy controls related to better patient-reported outcomes in sport and recreation. Similarly, we found that a decrease in LyE $\lambda_{L}$ was also somewhat correlated with a decrease in symptoms.

Lastly, increases in sample entropy values in the ML direction were positively correlated with increased patient-reported quality of life. Similarly to our short-term stability measures, an increase in sample entropy reflected a trend towards values found in our asymptomatic cohort. All correlation coefficients and significant values can be found in Table 4.

## 4. Discussion

Based on this study’s findings, a single trunk worn accelerometer can be used to assess gait quality in people with OA. The parameters of interest between the people with OA and the asymptomatic controls are symmetry and stability. Furthermore, the results show that symmetry, stability, and complexity are of interest to monitor in the follow-up of people treated with a TKA. Large inter-individual differences can be observed within the violin plots in pre-TKA and post-TKA. These differences highlight the need for personalised rehabilitation trajectories. As such, using movement quality parameters derived from a single accelerometer could possibly be used to fine-tune a person’s rehabilitation trajectory. These results are a step in the direction towards data-informed rehabilitation strategies.

To reach our first aim, we found that of all studied parameters, we found a more asymmetric gait pattern in people with hip OA as well as a more asymmetrical and rigid gait pattern in people with knee OA compared to asymptomatic controls. However, only the symmetry but not the stability surpassed the minimal detectable change, indicating that the groups’ differences surpass naturally occurring between-session variation and measurement error. Therefore, we can confidently state that symmetry measures can detect differences in movement quality between people with OA and asymptomatic controls.

Gait symmetry was significantly lower in hip and knee OA patients than in asymptomatic individuals. These results corroborate previous research finding gait asymmetry in people with unilateral OA [42,43,44]. This compensatory gait pattern (i.e., more asymmetry) might also lead to a higher risk of developing OA in other joints, as aberrant mechanical loading is a risk factor for the onset and progression of OA [5].

Short-term dynamic stability (LyE) was not significantly affected in people with hip OA, whereas, in people with knee OA short-term dynamic stability values in all three directions were significantly lower than asymptomatic controls. This finding was surprising since higher maximum divergence exponents during walking have been associated with increased fall risk [36]. However, when we consider stability in having an optimum (e.g., healthy people), both significantly higher and significantly lower extremes are undesirable. The maximum Lyapunov Exponent is calculated as the logarithmic rate of divergence of initially nearest neighbours in an attractor state, in our case, steady-state walking. In one extreme, when the LyE is significantly increased, the system can be considered unstable and is less able to handle perturbations (i.e., a small perturbation will knock the system off course), i.e., an undesirable outcome. Alternatively, when the LyE is significantly smaller, it might reflect a too rigid system. It might reflect limited adaptability to change motor patterns (e.g., changing from straight-line walking to sidestepping an unanticipated obstacle). Considering we found lower LyE values in the people with knee OA, we theorise that people with knee OA adopt a more rigid movement pattern to ensure stability. Similarly, in people with anterior cruciate ligament deficient knees a lower LyE was found during backward walking than in healthy controls [45]. Likewise, they appointed these lower values to rigidity in the motor pattern. So far, the maximum Lyapunov Exponent has been investigated in people with known fall risk or patients with focal cerebellar lesions [36,46]. A higher Lyapunov exponent is expected within those populations. However, based on our findings and those by Zampeli et al. (2010) [45], it might be relevant to investigate this parameter in populations whose gait pattern is expected to be more rigid, e.g., Parkinson’s, walking on ice, or chronic pain. However, we need to investigate this phenomenon and its consequences in more detail in future research.

Unlike gait stability and symmetry, gait complexity and smoothness did not significantly differ between either hip or knee OA and asymptomatic controls. This result means that the gait pattern of people with OA is still quantified by a healthy physiological, biomechanical pattern and indicates well-trained motor behaviour.

To evaluate the sensitivity of the movement quality parameters to capture gait changes during the four re-evaluation moments post-TKA, we found that by using a single trunk worn IMU, changes in symmetry, stability and complexity can be detected. Following TKA, we observe an initial deterioration in symmetry, followed by a significant improvement after six months. These significant improvements in symmetry resulted in a normalisation of these values—at 12 months post-TKA, the group was no longer significantly different from asymptomatic controls.

We observed a similar pattern in short-term dynamic stability, an initial (non-significant) further decrease followed by an increase at 6 and 12 months. However, only in the vertical direction, at 12-months post-TKA, there is no longer a significant difference with asymptomatic controls. Additionally, we found a gradual increase in the long-term stability; this increase was too small to be significantly different between re-evaluation moments. However, it resulted in a significantly larger long-term LyE in people 12 months post-TKA than in asymptomatic controls, reflecting more gait pattern fluctuations post-TKA Even though only the short-term LyE was associated with the risk of falling [36], Su and Dingwell (2007) argued that although the long-term maximum Lyapunov Exponent did not predict fall risk, it does quantify inherent stability [47]. Furthermore, Terrier et al. (2018) showed that this measure reflects the correlation between the different strides, with higher values indicating less correlation and more fluctuations [35]). Therefore, from our results, we theorise that over a time increment of a single step, people post-TKA adopt a more rigid movement pattern to ensure stability; however, that might compromise the dynamic stability over the time increment of several strides by limiting the adaptability of the system. Future studies will need to investigate whether this theory can be confirmed.

Furthermore, we observe an initial decrease in movement complexity (sample entropy) at six weeks post-TKA that shows a trend toward normalisation after six months. A healthy gait pattern is characterised by a high degree of complexity to adapt to unpredictable events [17,48]. Therefore, the drop in movement complexity leads to a gait pattern that is less adaptable to unexpected changes and is less able to use step-to-step adjustments to regulate balance control effectively [17,49], which, in turn, could lead to an increased risk of falling. Contrastingly, movement smoothness did not seem to be affected by the TKA as we observed no changes or trends between the timepoints. We, therefore, conclude that movement smoothness quantified by the log dimensionless jerk is not sensitive to biomechanical gait changes.

The study’s third aim was to investigate whether changes in movement quality parameters between 6 weeks and 12 months following TKA are related to patient-reported functioning, pain, symptoms, sports/recreation, and quality of life. Based on the results, we found some fair to moderate correlations between changes in movement quality and changes in patient-reported functioning.

The most notable is the significant correlations between stability and patient-reported pain, quality of life, performing activities of daily living, and sports/recreation. In fact, a larger increase in short-term LyE in ML and AP direction from 6 weeks to 12 months post-TKA (more towards values reported in asymptomatic controls) was correlated with a larger recovery in patient-reported functioning in daily life activities, less hinder from pain and a better quality of life. The quality-of-life section of the KOOS mainly reflects how well people can “trust” their affected knee (i.e., giving away). Thus, changes in objectively measured stability appear to correlate with changes in subjective feelings of being stable. Similarly, a more considerable increase in sample entropy in ML direction was related to a more considerable improvement in patient-reported quality of life; i.e., higher complexity relates to a better quality of life. This correlation could reflect the hypothesis that ML motion is under a direct feedback control loop for step-by-step adjustments for effective balance control [49]. Hence, higher complexity (i.e., towards more healthy values) might be related to better balance control and relate to a better quality of life.

All our findings combined—i.e., the initial decrease in quality followed by a recovery and normalisation—show the potential of a single accelerometer to follow up the recovery post-TKA. The correlations found between improvements in symmetry, stability and complexity measures and improvements in patient-reported outcomes could identify objective treatment targets (e.g., increase stability) to improve patient-reported functioning. Future work should determine if incorporating these objective measures in clinical practice ensures a better and more personalised rehabilitation plan that improves functioning.

There are some limitations to this work. Foremost, the sample entropy and the maximum Lyapunov Exponent are sensitive to the number of data points used. We tried to overcome this by including the same number of steps for each individual. However, this does indicate that the absolute number reported here cannot be compared to the values reported in the literature. TenBroek et al. (2007) showed that the LyE value increases with fewer data points and stabilises after 5000 samples [50]. Similarly, Yentes et al. (2012) proved that the sample entropy stabilises after 2000 samples [37]. In this study, the number of data points is around 1500 samples, below the thresholds for the LyE and sample entropy. Therefore, we most likely overestimated the absolute values of the LyE and sample entropy. However, because we used the same number of steps for each individual, we believe that we can compare the values between our different groups, just not with previously reported work. We could solve the limited number of samples problem using an accelerometer with a higher sampling frequency. Most likely, this will change the absolute values of the parameters. Since sample entropy quantifies waveform (un)predictability, using a higher sampling frequency might be more valid. Using relatively low frequencies, we might not have captured the actual complexity of the signal. These frequency-specific limitations make comparing our results to previous literature and other accelerometers with different specifications complex. As such, the findings of this study are specific to the type and placement of the accelerometer.

Furthermore, we used straight-line walking within a controlled lab-based setting. Considering we did not use lab-based measurement equipment, we can directly translate these methods to a clinical setting (i.e., only an accelerometer and long hallway are needed). While we found significant differences between our groups and following TKA during walking, these methods might be even more sensitive in more complex tasks like ascending or descending stairs or turning. Future research could incorporate more complex daily life activities to gain more insights into symmetry, stability, complexity, and smoothness during daily life activities in people with OA and following TKA. Furthermore, we only included a limited number of subjects in this study. Considering that this is a secondary analysis of a more extensive study, the sample size was based on compartmental forces of the knee, which are not comparable to the parameters used within this work. Post hoc sample size calculation did show that we were slightly underpowered for sample entropy and the maximum Lyapunov Exponent to distinguish the gait pattern of hip OA people from asymptomatic controls. Recognising the ease with which these parameters can be collected, future research should include more subjects. In combination with incorporating these movement quality parameters in rehabilitation, to ensure that data informed personalised rehabilitation targets lead to an improved function in people with either hip or knee OA and following total knee arthroplasty.

## 5. Conclusions

A single lower back accelerometer can be used to characterise movement quality before and after a total joint arthroplasty without the restrictions of a gait lab. We found a more asymmetric gait pattern in people with hip OA as well as a more asymmetrical and rigid gait pattern in people with knee OA compared to asymptomatic controls. Concluding that symmetry and stability are measures of interest in people with OA. We also found that symmetry, stability, and complexity are sensitive to biomechanical changes post-TKA, showing the ability to objectively monitor time-sensitive changes in movement quality after a total joint arthroplasty. Furthermore, the correlations found between improvements in symmetry, stability and complexity measures and improvements in patient-reported outcomes might identify objective treatment targets (e.g., increased stability) to improve patient-reported functioning. Future work should determine if incorporating these objective measures in clinical practice ensures that data-driven personalised rehabilitation leads to improved functioning of people with (either hip or knee) OA and after total knee arthroplasty.

## Figures and Tables

**Figure 1 sensors-22-02955-f001:**
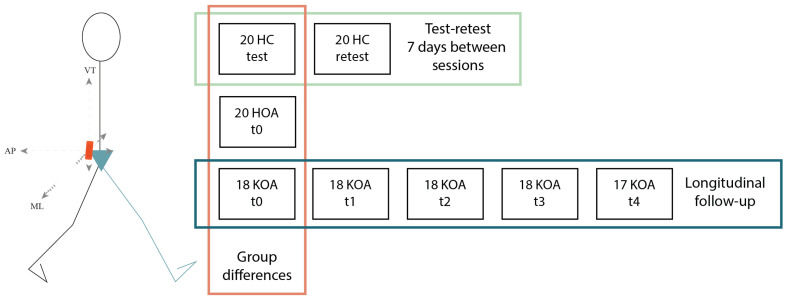
Sensor setup and data collection flow. Enclosed by the red box is the test–retest reliability data, the yellow box is the longitudinal follow-up data, and the blue box represents the cohort data. HC = asymptomatic control, HOA = hip OA, KOA = knee OA.

**Figure 2 sensors-22-02955-f002:**
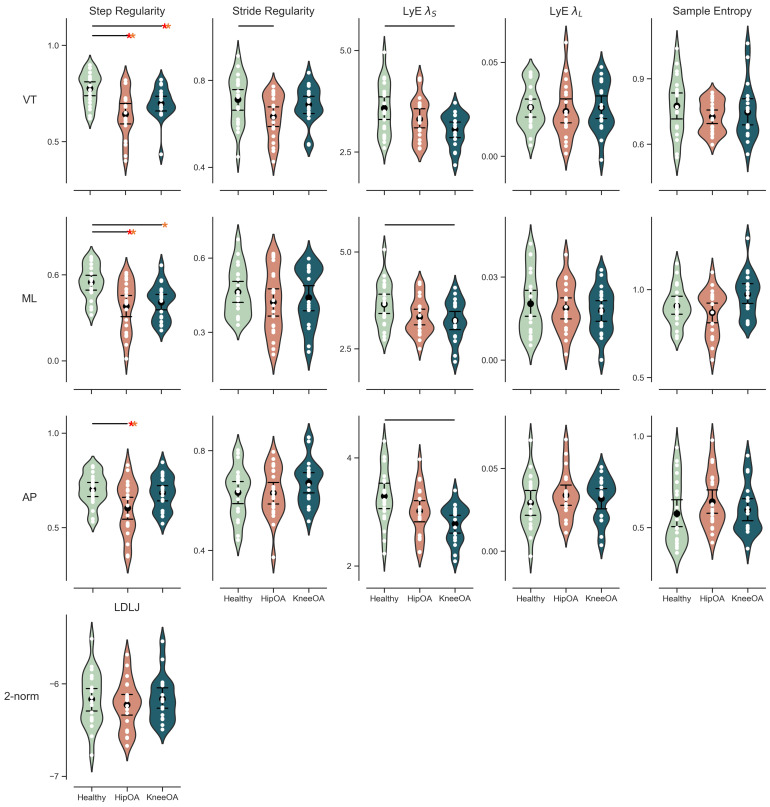
Violin plots of the group differences–healthy in green, hip OA in orange, and knee OA in blue–of step/stride regularity, short- and long-term divergence exponents, sample entropy in three directions, and smoothness as a vector. The white dots represent an individual measurement; the black dot is the mean with the corresponding confidence interval. Black lines above the violins indicate a significant difference between the groups. The red and orange asterisk indicates that the difference exceeds the minimal detectable change (MDC) and 95% confidence interval.

**Figure 3 sensors-22-02955-f003:**
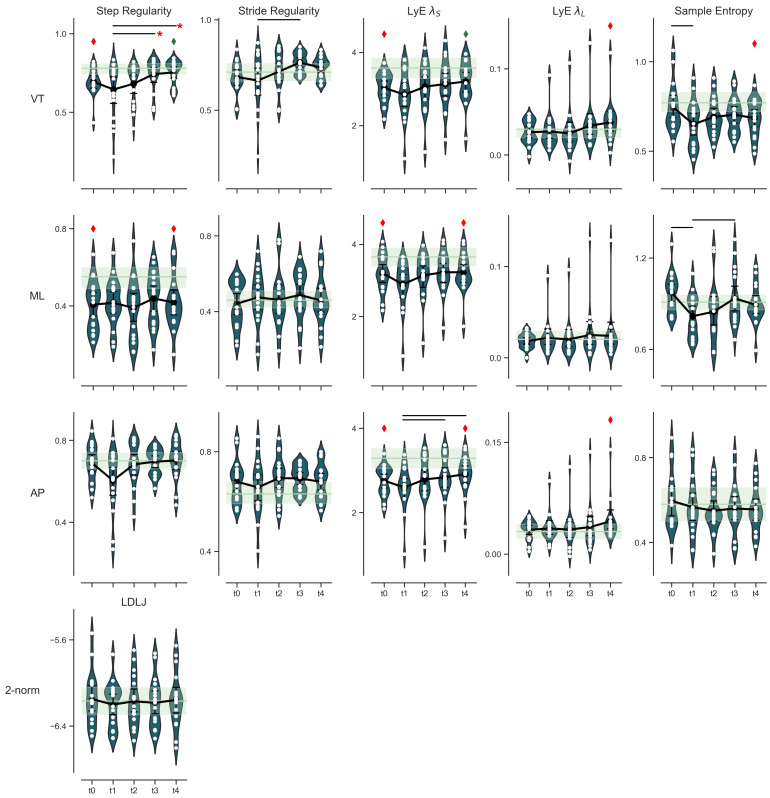
Violin plots of the longitudinal follow-up of gait symmetry, step/stride regularity, short- and long-term divergence exponents, sample entropy in three direction, and smoothness as a vector post total knee arthroplasty. Pre-TKA (t0), six weeks, three months, six months and 12 months (t1–t4, respectively). The white dots represent an individual measurement; the black dot is the mean with the corresponding confidence interval. Black lines above the violins indicate a significant difference between the time points. The red asterisk indicates that the difference exceeds the minimal detectable change (MDC). The green shaded bar shows the mean plus confidence interval of the healthy control subjects. A red diamond indicates that that time point was significantly different from the healthy controls. Green diamond shows that that difference is no longer significant between 12 months post-TKA and healthy individuals (TKA).

**Table 1 sensors-22-02955-t001:** Inclusion and exclusion criteria for the participants.

	Healthy Population	Patient Population
Inclusion	Aged between 50–75 years old Understand the Dutch language Able to walk 10 m Able to ascent/descent the stairs	Aged between 50–75 years old Understand the Dutch language Diagnosed with hip or knee OA Awaiting total hip or knee replacement surgery Able to walk 10 m Able to ascent/descent the stairs
Exclusion	Diagnosed with musculoskeletal or neurological disorders Pain in hips, knees or ankles that affect normal movement	Corticosteroid injection 3 months before inclusion to the study Diagnosed with symptomatic hip or knee OA on the contralateral knee Joint replacement in other lower limb joints Symptomatic degenerative disorders in other lower limb joints Neurological conditions that could alter movement pattern History of pathological osteoporotic fractures (in hip, knee or ankle joints)

**Table 2 sensors-22-02955-t002:** Participant characteristics mean (SD).

	Control	Hip OA	Knee OA	Knee OA	Knee OA	Knee OA	Knee OA
	t0	t0 ^1^	t0 ^1^	t1 ^1^	t2 ^1^	t3 ^1^	t4 ^1^
Mass (kg)	70.8 (14.2)	75.4 (11.6)	79.8 (8.2)	79.8 (8.5)			
Height (m)	1.70 (0.08)	1.75 (0.09)	1.75 (0.08)	1.75 (0.08)			
Age (years)	62.7 (8.5)	63.1 (6.2)	65.1 (5.1)	64.7 (4.9)			
Sex (M/F)	9/11	11/9	11/7	10/7			
PROM ^1^							
Pain	95.35 (6.1)	50.85 (11.85)	49.77 (13.89)	54.34 (14.89)	63.25 (16.04)	78.99 (16.52)	78.04 (14.43)
Symptoms	96.9 (5.5)	52.25 (17.9)	50.89 (22.90)	52.32 (12.02)	59.82 (13.97)	69.34 (16.00)	80.06 (13.83)
ADL	98.7 (2.55)	56.4 (15.45)	60.17 (17.92)	62.38 (14.03)	73.15 (16.13)	82.52 (14.94)	87.38 (12.66)
Sport	93.3 (9.2)	24.05 (23.2)	29.44 (26.64)	18.96 (21.25)	31.18 (16.65)	47.81 (25.32)	55.31 (27.00)
QOL	92.05 (9.8)	27.95 (15.85)	31.25 (17.06)	39.58 (17.34)	42.71 (15.27)	57.29 (16.61)	61.98 (17.16)

^1^ t0 = pre-THA or pre-TKA, t1 = 6 weeks post-TKA, t2 = 3 months post-TKA, t3 = 6 months post-TKA, t4 = 12 months
post-TKA, PROM = Patient-reported outcome measures, ADL = Activities of daily life, QOL = Quality of life.

**Table 3 sensors-22-02955-t003:** Test statistics group comparison. Averages of the test (avg t0) and retest (avg t1) session of the asymptomatic controls with the associated interclass correlation coefficient (ICC) and the minimal detectable change (MDC). The averages for the hip OA and knee OA subjects are reported, with the difference in mean (Diff) between the healthy cohort and an OA cohort. All significant *p*-values (<0.05) and differences that surpass the MDC are in bold.

	Asymptomatic Controls				Hip OA			Knee OA		
	avg t0	avg t1	ICC	MDC	avg t0	Diff	*p*-Value	avg t0	Diff	*p*-Value
VT										
StepRegularity	0.78	0.77	0.89	0.05	0.64	**0.13**	**<0.001**	0.70	**0.08**	**<0.01**
StrideRegularity	0.71	0.71	0.79	0.12	0.63	0.08	**0.02**	0.69	0.02	0.23
SampEn	0.77	0.85	0.75	0.20	0.73	0.05	0.08	0.75	0.03	0.21
LyE $\lambda_{S}$	3.57	3.47	0.70	0.85	3.31	0.27	0.09	3.05	0.52	**<0.01**
LyE $\lambda_{L}$	0.03	0.03	0.02	0.06	0.02	0.00	0.26	0.03	0.00	0.45
ML										
StepRegularity	0.55	0.55	0.79	0.15	0.38	**0.17**	**<0.01**	0.41	0.14	**<0.001**
StrideRegularity	0.46	0.48	0.49	0.27	0.42	0.04	0.16	0.44	0.02	0.31
SampEn	0.91	0.95	0.80	0.13	0.87	0.04	0.31	0.97	0.07	0.07
LyE $\lambda_{S}$	3.65	3.67	0.80	0.56	3.32	0.33	**0.04**	3.23	0.42	**0.02**
LyE $\lambda_{L}$	0.02	0.02	−0.41	0.07	0.02	0.00	0.46	0.02	0.00	0.37
AP										
StepRegularity	0.70	0.70	0.87	0.05	0.60	**0.10**	**0.02**	0.68	0.02	0.28
StrideRegularity	0.63	0.61	0.78	0.11	0.63	0.00	0.48	0.67	0.04	0.12
SampEn	0.58	0.60	0.91	0.09	0.64	0.06	0.06	0.60	0.02	0.20
LyE $\lambda_{S}$	3.29	3.28	0.86	0.38	3.01	0.27	0.05	2.78	**0.51**	**<0.01**
LyE $\lambda_{L}$	0.03	0.03	−0.86	0.10	0.03	0.00	0.20	0.03	0.00	0.29
Norm vector										
LDLJ	−6.17	−6.10	0.90	0.14	−6.23	0.06	0.24	−6.16	0.00	0.48

**Table 4 sensors-22-02955-t004:** Correlations between movement quality parameters and patient-reported functioning. Moderate and significant correlations are in bold.

	Pain		ADL		QOL		Symptoms		Sports/Rec	
	ρ	*p*-Value	ρ	*p*-Value	ρ	*p*-Value	ρ	*p*-Value	ρ	*p*-Value
VT										
StepRegularity	−0.15	0.55	−0.28	0.28	0.06	0.83	−0.32	0.21	−0.40	0.13
StrideRegularity	−0.11	0.69	0.05	0.84	0.16	0.55	−0.01	0.96	−0.16	0.57
SampEn	0.19	0.46	0.18	0.49	0.09	0.74	0.28	0.27	−0.16	0.57
LyE $\lambda_{S}$	0.34	0.19	0.34	0.18	0.38	0.14	0.16	0.55	0.05	0.86
LyE $\lambda_{L}$	−0.08	0.76	−0.04	0.87	−0.14	0.59	**—0.43**	0.09	−0.08	0.77
ML										
StepRegularity	−0.00	0.99	0.07	0.78	0.09	0.74	−0.01	0.98	−0.27	0.32
StrideRegularity	0.04	0.89	0.18	0.50	0.08	0.77	0.27	0.29	−0.25	0.37
SampEn	0.11	0.69	0.05	0.85	**0.47**	0.05	0.24	0.36	0.09	0.76
LyE $\lambda_{S}$	**0.51**	**0.03**	**0.43**	0.08	**0.43**	0.09	0.15	0.56	−0.05	0.85
LyE $\lambda_{L}$	−0.10	0.71	−0.02	0.93	−0.39	0.12	−0.17	0.52	**—0.56**	**0.03**
AP										
StepRegularity	0.28	0.28	0.26	0.32	**0.48**	0.05	−0.08	0.76	−0.05	0.85
StrideRegularity	−0.06	0.82	0.19	0.46	0.15	0.58	0.11	0.66	−0.26	0.34
SampEn	0.00	0.99	−0.27	0.30	−0.22	0.40	−0.23	0.37	−0.09	0.75
LyE $\lambda_{S}$	**0.54**	**0.03**	**0.55**	**0.02**	**0.48**	0.05	0.28	0.28	0.11	0.69
LyE $\lambda_{L}$	−0.17	0.52	0.13	0.62	−0.07	0.78	−0.10	0.69	**—0.40**	0.14
2-norm										
LDLJ	−0.05	0.85	0.06	0.82	0.10	0.71	0.15	0.57	**—0.51**	0.05

## Data Availability

The data that support the findings of this study are not openly available due to GDPR guidelines (i.e., human motion data), and are available from the corresponding author upon reasonable request.

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
