# Peer review of "Movement Quality Parameters during Gait Assessed by a Single Accelerometer in Subjects with Osteoarthritis and Following Total Joint Arthroplasty"

_sensors, 2022, doi:10.3390/s22082955_

Round 1

Reviewer 1 Report

-The abstract can be majorly improved to read more concisely. The numbering system used in the description can be expressed in a better manner.

-It is suggested to replace the word ‘moreover’ with ‘furthermore’.

-The background theory of the IMU in gait analysis can be strengthened to include more examples. Especially emphasizing the limitations and future work proposed by other studies.

-Table formats should conform to MDPI standards.

-The coordinate conversions in 2.3 should be detailed more clearly.

-It is strongly recommended that a figure of the system proposed be included in the 2nd section.

-A brief explanation of the Lyapunov Exponent would improve the section.

-Please explain why the Friedman’s chi-square ANOVA test was used.

-The structure and explanation of the violin plots can be majorly improved. Instead of having 1 figure per page with a long caption, consider having a shorter caption and explaining each plot in text. This will help reduce page length and make the discussion more concise.

-You mention an interesting point in lines 267-270 correlating the step symmetry with quality of life. It would be interesting to hear more about this in detail in that section. How were patients’ quality of life evaluated? What kind of questions were asked?

-There is no need to repeat the aims for a 3rd time at the start of the discussion.

-You should also address the many limitations of using accelerometers and IMUs, especially in high precision applications such as gait analysis. Adding a small section at the end of the discussion describing this should be considered.

-It would be good to address the violin plots directly in the discussion section also as they depict interesting results.

-From a system design perspective, it would have been interesting to have compared a single trunk-worn IMU with another IMU setup, such as attaching IMU to the head, lower back or legs. Then compare the results of both systems. Perhaps this could be future work.

-The discussion in lines 316-330 are key findings and should be extended to include more detail. The results regarding the short-term dynamic stability are particularly interesting.

-Overall the English level and grammar in the paper can be significantly improved, It is strongly suggested that the authors use an English-checking service before resubmitting.

Reviewer 2 Report

This manuscript contributes to further research on the effects of receiving TKA treatment on gait symmetry, stability, and complexity. Although I am satisfied with the analytical part of the manuscript, I strongly recommend adding a clear description of the technical part to the paper. Here are some suggestions to consider implementing in your next version. 1.Lack of description of the experimental setup. 2. Lack of pictures to support the experiment. For example, I suggest adding some pictures of real scenes wearing IMU to assist in illustrating the section of data acquisition. 3.A specific description of the method is missing in the text. How the tilt of the sensor is corrected and how the coordinate systems are transformed. 4. Since some established techniques were used in the paper, did the authors adapt these original techniques and then apply them to their own problems. 5. What does the number of red asterisks in Figure 2 represent?

Round 2

Reviewer 2 Report

This is an significant study on gait assessment in subjects with osteoarthristis based on wearable sensors systems. The paper is well organized with clear figures and supporting charts. It is well written and the content is comprehensive.  Overall, I agree to publish this work in Sensors.